# SRC and ERK Regulate the Turnover of Cytoskeletal Keratin Filaments

**DOI:** 10.3390/ijms26125476

**Published:** 2025-06-07

**Authors:** Marcin Moch, Rudolf E. Leube

**Affiliations:** Institute of Molecular and Cellular Anatomy, RWTH Aachen University, 52074 Aachen, Germany

**Keywords:** EGFR, SRC, FAK, PI3K, AKT (PKB), ERK1/2 (MAPK3/1), keratin, intermediate filaments, cytoskeleton, A431, FRAP

## Abstract

Epithelial differentiation and function are tightly coupled to the keratin intermediate filament cytoskeleton. Keratin filaments are unique among the cytoskeletal filament systems in terms of biochemical properties, diversity and turnover mechanisms supporting epithelial plasticity in response to a multitude of environmental cues. Epidermal growth factor (EGF) is such a cue. It is not only intricately intertwined with epithelial physiology but also modulates keratin filament network organization by increasing keratin filament turnover. The involved EGF receptor (EGFR)-dependent intracellular signaling cascades, however, have not been identified to date. We therefore tested the effect of selective inhibitors of downstream effectors of the EGFR on keratin filament turnover using quantitative fluorescence recovery after photobleaching experiments as readouts. We find that SRC and ERK kinases are involved in the regulation of keratin filament turnover, whereas PI3K/AKT and FAK have little or no effect. The identification of SRC and ERK as major keratin filament regulators extends beyond EGF signaling since they are also activated by other signals and stresses. Our data unveil a mechanism that allows modification of the properties of keratin filaments at very high temporal and spatial acuity.

## 1. Introduction

The epithelial cytoskeleton consists of three structurally and compositionally distinct networks composed of either actin filaments, microtubules or keratin intermediate filaments. The epithelium-specific keratin filament system differs fundamentally from the other ubiquitous systems in terms of biophysical properties, morphogenesis and function. It is also the least understood. For example, the assembly and disassembly mechanisms of keratin filaments are only in part characterized. It has been proposed that the keratin network undergoes a continuous cycle of assembly and disassembly in cultured cells [1,2,3,4]. This turnover cycle originates typically in the cell periphery, where small keratin particles appear that elongate and move towards the cell center. The growing particles are subsequently integrated into the keratin filament network, which translocates to the nucleus. Filaments then either form a stable cage-like structure around the nucleus or disassemble into soluble subunits, which diffuse rapidly and can be re-utilized for another round of keratin filament assembly. This turnover cycle is complemented by lateral exchange of subunits within the keratin filament network [2,5].

Even less is known about the regulation of keratin turnover. There is consensus, however, that post-translational modifications are involved [6,7,8]. They directly affect filament stability through changes in subunit exchange and bundling [6,9,10]. Alternatively, filament stability is regulated through association with other proteins such as plakin domain-containing proteins, heat shock proteins and signaling proteins [11,12,13,14,15]. How and in which way these diverse processes are regulated remains to be studied. It has been suggested that key signaling pathways are involved including epidermal growth factor receptor (EGFR), cellular sarcoma kinases (SRCs) and mitogen-activated protein kinases (MAPKs) [16]. It has been shown repeatedly that activation of these pathways results in altered keratin phosphorylation, which can affect the solubility of keratin subunits [9,17,18,19,20,21,22,23,24,25]. In addition to these observations, there is limited direct evidence connecting pathway activation with keratin network dynamics. Time lapse fluorescence microscopy showed that phosphatase inhibition induces keratin filament network disassembly into keratin granules, which could be prevented by p38 MAPK inhibition [26,27,28]. Wöll et. al. [9] further provided direct evidence for the involvement of p38 MAPK in rapid keratin network disassembly coinciding with keratin phosphorylation. A much more subtle effect was observed by Moch et al. 2013 [18], who showed that EGFR stimulation and inhibition in serum-starved cells increased or decreased keratin network turnover, respectively, as assessed by bulk flow and fluorescence recovery after photobleaching (FRAP) experiments. These findings exemplify how growth factors and their associated downstream signaling pathways regulate the plasticity and turnover of the keratin filament system.

The goal of the current study was to identify such signaling pathways that may be involved in the regulation of keratin filament turnover.

## 2. Results

### 2.1. Experimental Design

The well-established vulvar carcinoma-derived A431 cell line was chosen for the experiments since it has served to characterize the keratin cytoskeleton (e.g., [18,29,30]) and EGF signaling [31,32]. A431 cells have an amplification of the EGFR gene leading to increased basal EGFR activation, making this cell line an ideal model for investigating downstream signaling in a serum free environment without the addition of growth factors. This allowed us to perform experiments in serum-starved cells without noticeable adverse effects on cell morphology. A drawback of this approach was that the growth factor-dependent pathways were not upregulated. Inhibition of those pathways was therefore expected to elicit only mild effects.

We used the previously described single cell clone AK13-1 that stably expresses green fluorescence protein-tagged human keratin 13 (HK13-EGFP in Windoffer and Leube, 1999 [29]; referred to as keratin 13-GFP in the current paper). Keratin 13 is one of the major keratin polypeptides in A431 cells. Previous research has demonstrated that it labels all keratin filaments in these cells due to its capability to pair with different partner keratin polypeptides [29,30]. As a result, we and others have used the AK13-1 cell line in multiple studies [12,18,26,27,28,33,34,35]. Cells were plated on laminin-332-rich matrix, which was obtained from squamous cell carcinoma-derived 804G cells to enhance spreading and adhesion for optimized fluorescence imaging. All experiments were performed 1 day after plating using at least two different pharmacological inhibitors for each target to minimize off-target effects. The time course and efficiency of inhibition was assessed by immunoblotting in each scenario.

We have previously shown that keratin filaments are continuously assembled in the periphery of epithelial cells from where they are transported towards the cell center to become part of the stable perinuclear cage or to be disassembled for a new round of assembly [4,18,36]. The assembly process is best seen in single cells without cell-cell connections and at the edge of cell colonies. FRAP analyses are well suited to capture filament dynamics. This method provides information on effective local filament turnover in defined regions of a given cell. The turnover is determined by multiple processes encompassing subunit diffusion, filament transport, filament formation and disassembly as well as subunit recruitment and release.

Figure 1A shows a representative example of the experimental FRAP setup used for the current study. The cell culture medium was exchanged for either control medium containing the solvent or medium containing the dissolved inhibitor of choice prior to imaging for either 5–10 min or 10–15 min depending on the onset of inhibitor action. Individual dishes were used for each FRAP experiment in which three identical regions of interest (ROIs) were defined in each of three cells for bleaching. The ROIs were always away from cell-cell contact areas at the periphery of colonies (blue dashed boxes in Figure 1A). For turnover analyses smaller ROIs (one quarter in size) were utilized that were within the bleached areas (white dashed boxes in Figure 1A) to exclude confounding effects of unbleached filaments moving into the respective ROIs during the recovery phase. Furthermore, the recording was performed at low laser intensity to prevent detectable bleaching.

In contrast to classical FRAP experiments, where rapid diffusion of fluorescence within the cytosol is measured immediately, fluorescence recovery measurements were initiated 1 min after bleaching. In this way, the effects of keratin subunit diffusion were minimized and small changes in keratin filament dynamics could be detected. To calculate the fluorescence turnover, the fluorescence measured 1 min after bleaching was subtracted from the fluorescence measured before and after bleaching within the small ROIs. Fluorescence gain was then compared at each time point to the pre-bleach fluorescence. To account for the considerable variability of keratin filament turnover (Figure 1A; Appendix A), each experiment series was performed on five different days (*n* = 5).

To determine the correct statistical model, we checked whether the FRAP measurements at the 10-min time points showed a normal or skewed distribution. As not all data sets exhibited a normal distribution, we opted to use medians rather than means and applied statistical non-parametric tests.

### 2.2. Inhibition of SRC Reduces Keratin Filament Turnover

Based on our previous observation that EGF signaling regulates keratin dynamics, we focused on SRC as an essential component of the EGFR downstream cascade [18].

PP 1 and SKI-606 were selected as well-characterized and specific SRC inhibitors [37,38]. To determine effective inhibitor levels, cells were treated with 1 to 50 µM PP 1 and 1 to 20 µM SKI-606 for 5 min prior to lysis and quantitative immunoblot analysis of tyrosine 416 SRC phosphorylation (Figure 1B; Appendix A). Next, 5 µM SKI-606 and 20 µM PP 1 were selected as effective concentrations for subsequent FRAP experiments.

Figure 1C displays graphs of the time-dependent fluorescence recovery revealing almost identical results for both inhibitors. After 10 min, the median turnover of keratin 13-GFP in control cells (35.7%) was twofold higher than in PP 1-treated cells (17.6%) and SKI-606-treated cells (16.7%). The decrease in keratin 13-GFP turnover was significant for PP 1 and SKI-606 (*p* = 0.032 and *p* = 0.014, respectively). Remarkably, the fluorescence recovery continued steadily in the control but appeared to reach a plateau after 10 min in the inhibitor-treated cells.

### 2.3. Inhibition of FAK Signaling Does Not Affect Keratin Turnover

It is known that the focal adhesion kinase (FAK) is an upstream regulator of SRC and is associated with EGFR signaling [39,40]. For example, EGFR activation can lead to integrin-mediated signaling events that involve FAK activation as part of cell migration [41,42]. We therefore wanted to test the effect of FAK inhibition on keratin turnover. To this end we selected well-characterized and specific FAK inhibitors PF-573228 [43] and PF-431396 [44].

To determine effective FAK inhibitor levels, cells were treated with 0.1, 0.5, 2.0 and 5 µM PF-573228 or PF-431396 for 5 min as shown in Figure 2A and Appendix A. For FRAP experiments, 0.5 µM PF-431396 and 2 µM PF-573,228 were selected. Both FAK inhibitors did not show a significant effect on keratin filament turnover (Figure 2B). The median turnover of keratin 13-GFP was similar after 10 min between control cells (25.0%), PF-431396-treated cells (26.4%) and PF-573228-treated cells (27.5%).

### 2.4. Inhibition of PI3K/AKT Signaling Has Only a Minor Effect on Keratin Turnover

Another important downstream pathway of the EGFR is phosphoinositide 3-kinase and protein kinase B (PI3K/AKT) signaling, which is considered to be independent of the SRC pathway although limited crosstalk has been reported [45,46]. AKT has more than 100 substrates and controls various biological responses, such as motility, metabolism, proliferation and cell growth [47]. Furthermore, keratins have been shown to bind AKT [48,49,50] with consequences for cellular resilience, proliferation and tumorigenesis (see also [51,52]).

The efficiency of PI3K inhibitors was assessed by measuring the phosphorylation levels of S473-AKT, which is activated downstream of PI3K. To make sure not to miss a delayed S473-AKT phosphorylation we monitored the inhibitor effects for up to 30 min (Figure 3A; Appendix A). PI 828 [53] induced near-maximal inhibition at 5 µM within 60 s, whereas wortmannin [54] required 5 min to achieve similar results at the same concentration.

In FRAP experiments shown in Figure 3B, PI 828 impacted keratin turnover significantly with a *p* value of 0.022, whereas wortmannin did not (*p* = 0.792). Of note, the median keratin turnover was lower for both inhibitors after 10 min of fluorescence recovery time (control = 28.0%; PI 828 = 17.0%; wortmannin = 19.9%).

Next, we wanted to find out whether AKT may directly affect keratin turnover. This was tested with Akti-1/2 [55] as shown in Figure 3C. The inhibitor demonstrated similar efficiency to wortmannin but at a lower concentration, i.e., 1 µM vs. 5 µM. As shown in Figure 3D, the treated cells had a lower median keratin turnover (control = 30.1%; Akti-1/2 = 22.5%), but the *p* value of 0.056 was not significant.

In conclusion, the data suggests that the PI3K/AKT pathway does not significantly affect keratin turnover although a trend is detectable that may become significant at later time points.

### 2.5. Inhibition of ERK Decreases Keratin Filament Turnover

As a third pathway that acts downstream of the EGFR, we investigated extracellular signal-regulated kinase 1 and 2 (ERK1/2, p44/42 MAPK, MAPK3/1) signaling. The efficiency of the ERK inhibitors FR 108204 [56] and SCH 772984 [57] was tested in a time-dependent manner to determine the optimal inhibition protocol. The inhibition process, which was assessed by T202 and Y204 dephosphorylation of ERK1/2, was comparatively slow (Figure 4A; Appendix A). FR 108204 achieved near-maximum dephosphorylation after 10 min at a concentration of 25 µM (upper part of Figure 4A). SCH 772984 first increased and subsequently decreased phosphorylation of T202 and Y204 at a concentration of 2 µM (lower part of Figure 4A). Consequently, we increased the preincubation times of the cells with both inhibitors to 10–15 min for the FRAP experiments.

FR 108204 and SCH 772984 both induced a significant decrease in keratin filament turnover with a *p* value of 0.008 (Figure 4B). The median keratin turnover in the control group was 32.2% after 10 min, which was markedly decreased in FR 108204-treated cells (17.2%). Similarly, cells treated with SCH 772984 also had a significant reduction in median keratin turnover (17.8% versus 28.1%).

In conclusion, the data show that ERK1/2 impacts significantly keratin turnover with identical *p* values of 0.0079 for both inhibitors.

## 3. Discussion

While keratin filaments are considered to be the most stable component of the epithelial cytoskeleton, examination of filament dynamics in living cells and organisms has revealed different modes by which keratin filament networks can rapidly adapt to local requirements. This plasticity is enabled not only by filament assembly and disassembly as is the case for actin filaments and microtubules, but also by subunit exchange of seemingly stable filaments. Kinase-dependent remodeling may therefore either impact the keratin filament turnover cycle by modulating keratin filament formation and growth or modify lateral keratin filament subunit exchange [4,16,58]. These different modes may help to modify filament properties at very high temporal and spatial precision. How this process is regulated is virtually unknown. The identification of upstream regulators of keratin filament turnover in the present study provides valuable insights into how this may be accomplished. Our observations confirm the importance of EGFR signaling for keratin filament dynamics [18,59] and add further details of this pathway for the overall organization of the keratin-desmosome-hemidesmosome scaffold [60,61]. The present report goes beyond the previous experiments by investigating possible downstream effectors of EGFR signaling, which are also modulated by other pathways that are activated, e.g., by keratinocyte growth factor (FGF-7), transforming growth factor β (TGF-β), insulin-like growth factor 1 (IGF-1) and hepatocyte growth factor (HGF) [62,63,64,65,66]. We identify SRC and ERK family kinases as important mediators while PI3K/AKT and FAK signaling appear to be less or not relevant. SRC and ERK kinases are major regulators of epithelial function (Table 1).

Thus, SRC kinases have been implicated in the remodeling of the epithelial cytoskeleton and junctions affecting cell migration and invasion [67,68,69,70]. At the molecular level, SRC is activated by phosphorylation of Y418 and dephosphorylation of Y530 downstream of the EGFR and integrin signaling [71,72]. Furthermore, SRC inhibition stabilizes cadherin mediated cell-cell adhesion and decreases integrin mediated cell-matrix adhesion [73,74]. In contrast, SRC activation increases the dynamics of adhesion complexes [42,75,76]. Furthermore, downregulation of SRC results in decreased actin dynamics [68,77,78]. Effects of SRC on the keratin cytoskeleton are less understood. While it has been shown that the binding of SRC to keratins prevents SRC activation [79,80], the consequences of this interaction for keratin filament stability have not been investigated.

Given the known effect of SRC signaling on focal adhesions and FAK activation [81,82] we had expected an impact of FAK inhibition on keratin filament turnover. The finding that FAK signaling is not relevant for keratin filament turnover, however, shows that the coordinated appearance of focal adhesions and keratin particles [83,84] is not mediated through this kinase and is in line with previous reports that showed that intermediate filaments, notably vimentin [85,86,87,88] and synemin [89,90] but also keratins [79,91], are upstream of FAK.

The serine/threonine-specific, stress-activated MAPKs are subdivided into three groups: p38 MAPK, c-Jun N-terminal kinases (JNKs) and ERKs. They are central players in epithelial function [92,93]. MAPKs are induced by mechanical stress, osmotic challenges and irradiation and are linked to migration, wounding and inflammation [94,95]. Their involvement in keratin filament network regulation has been the topic of intense research efforts [3,16]. Several studies examined the effect of p38 MAPK on keratin filament network organization and keratin phosphorylation [9,19,21,22,23]. Of note, p38 MAPK have also been linked to desmosomal signaling with consequences for cytoskeletal organization and disease implications [96,97]. It is generally assumed that p38 MAPK and EGFR often participate in parallel or intersecting pathways [98]. On the other hand, JNKs are activated downstream of EGFR through intermediates such as RAS, MEK and ERK [92]. Interestingly, the induction of JNKs results in increased keratin phosphorylation [21,24,99]. Similarly, ERK activity also affects keratin phosphorylation [21,25] and is altered in keratinocytes with mutant keratins [100]. Furthermore, the EGFR-dependent regulators of keratin filament turnover identified in the current study have been linked to phosphorylation of specific keratin residues. For example, it has been shown that SRC activation leads to phosphorylation of K19-Y391 [20] and of K17-S44 [101] and that ERK1/2 induces phosphorylation of K8-S73, K8-S431 and K18-S52 [17,25,102]. These observations are in line with the EGF-dependent phosphorylation of K8-S73 and K8-S431 [17,18]. Keratin phosphorylation at these and other residues is known to increase the soluble keratin pool (e.g., [23,27,103]). The precise effector mechanism is less well understood [16]. It is likely, however, that the changes in keratin filament turnover detected in the current study contribute to the shift in keratin solubility.

Important mediators in the shift of keratin filament turnover are associated proteins. The best studied example in this context is probably epiplakin [33]. Binding of epiplakin to keratins was shown to be induced by different types of stress in a cytoplasmic calcium-dependent fashion. The association reduced keratin flow. How it affects keratin filament turnover, however, remains to be shown. It will be interesting to determine whether desmoplakin has a similar effect on keratin filament stabilization and thereby favors the reported desmosome-dependent keratin filament assembly [104].

The uncovered linkage between signaling pathways and keratin filament plasticity suggests that regulation of keratin filament turnover plays an important role in epithelial physiology by supporting increased cellular deformability to enable migration during wound healing and tumor metastasis. Our observations complement the large number of studies (reviews in [7,16,105]), which characterized downstream effectors of keratins including the same signaling pathways that we have identified in the current study to regulate keratin turnover. A complete understanding therefore requires a combined view of keratin filament regulators and effectors as parts of interdependent feedback cycles.

## 4. Materials and Methods

### 4.1. Cell Culture

A431 vulvar carcinoma derived subclone AK13-1 [29] was cultured in Dulbecco’s Modified Eagle’s Medium (DMEM) supplemented with l-alanyl-glutamine (Sigma-Aldrich, St. Louis, MO, USA) and 10% (*v*/*v*) SeraPlus fetal bovine serum (FBS) (PAN Biotech, Aidenbach, Germany) at 37 °C and 5% CO_2_. Cells that had reached confluency for 1–2 days were passaged twice weekly and seeded at a concentration of 10,000–20,000 cells/cm^2^ in 25 cm^2^ cell culture flasks with 6 mL cell culture medium (Greiner Bio-One, Frickenhausen, Germany). Therefore, cells were washed with phosphate-buffered saline (PBS) without Ca^2+^/Mg^2+^ (Sigma-Aldrich) and subsequently incubated with 0.05% trypsin (Genaxxon Bioscience, Ulm, Germany) at 37 °C for 5 min.

For experimental procedures, cells were plated on surfaces pre-coated with laminin 332-rich matrix derived from 804G rat cells [18,106]. Following an incubation period of 20 min in standard cell culture medium, the cells were washed with FBS-free DMEM and subsequently cultured in the same medium. For FRAP experiments cells were grown in 35-mm diameter glass bottom dishes (12 mm glass diameter, thickness 1.5#; MatTek, Bratislava, Slovakia) at a concentration of 35,000 cells/cm^2^. For immunoblot analysis cells were grown in 60 mm or 100 mm diameter cell culture petri dishes (Greiner Bio-One) at a concentration of 100,000 cells/cm^2^ for 24 h. FRAP experiments were conducted in 25 mM 4-(2-hydroxyethyl)-1-piperazineethanesulfonic acid-buffered DMEM without phenol red (Life Technologies, Carlsbad, CA, USA) and without FBS. Unless otherwise specified, cells were preincubated for 5–10 min with either drug-containing or control medium prior to FRAP analysis.

### 4.2. Pharmacological Inhibitors

Selective SRC family kinase inhibitor PP 1, dual SRC-ABL inhibitor Bosutinib (SKI-606), selective FAK inhibitor PF-573228, dual FAK/PYK2 inhibitor PF-431396, PI 3-kinase inhibitor PI 828, irreversible PI 3-kinase inhibitor wortmannin, selective dual Akt1/2 inhibitor Akti-1/2 and selective ERK inhibitor FR 180204 were obtained from Bio-Techne GmbH (Wiesbaden-Nordenstadt, Germany). The ERK1/2 inhibitor SCH 772984 was obtained from Cayman Chemicals (Ann Arbor, MI, USA).

All inhibitors were dissolved in pure dimethyl sulfoxide (DMSO; Sigma-Aldrich), aliquoted, flash-frozen in liquid nitrogen and stored at −80 °C. After use, the aliquots were discarded. The final concentration of DMSO was consistently 0.5% (*v*/*v*) in drug-treated samples and controls.

### 4.3. Microscopy

Microscopical recordings were conducted using a laser scanning confocal microscope (LSM 710) with Zen black 2.1 SP3 software (Carl Zeiss, Jena, Germany). The microscope was equipped with an Airyscan detector, oil immersion objective (63×/1.40-N.A., DIC M27) and a focus-shift correction system (DefiniteFocus; all by Carl Zeiss). For live-cell imaging, the microscope was maintained at 37 °C in an incubation chamber with 5% CO_2_.

Fluorescence recovery after photobleaching (FRAP) experiments were recorded at 60 s intervals before and after bleaching with the Airyscan detector set to standard confocal mode. The images (*x* = *y* = 89.97 µm = 1024 pixel) were acquired at a pixel dwell speed of 3.14 µs, gain 800 and 16 bit-depth in three dimensions at a *z*-resolution of 0.75 µm. The pinhole was set to 106 µm which corresponds to 2 Airy units for the used 460–480 and 495–550 nm emission range dual filter. For excitation of GFP the argon-ion laser (module LGK 7872 ML8; Carl Zeiss) was used at 488 nm and 0.1 to 0.2% power. In addition, the photon multiplayer was increased to the limit of safe operation. Bleaching was triggered automatically 60 s after the first *z*-stack within three areas of 14.06 × 10.54 µm (160 × 120 pixel) each. The bleaching process took 30 s, resulting in a 90-s interval between the image taken before bleaching and the one taken directly after bleaching. For bleaching the laser was set to 100% power and the target areas were rescanned 5–7 times at a dwell time of 6.28 µs. Please note that the increased scanning time or laser power was implemented to compensate for laser aging (within 2 years).

### 4.4. Antibodies and Immunoblotting

Primary monoclonal murine antibodies against SRC (dilution: 1:3000; clone: L4A1; Lot: 10) were from Cell Signaling Technologies (Danvers, MA, USA) and against FAK (1:3000; 4.47; 1970370) from Millipore (Burlington, MA, USA). Primary monoclonal rabbit antibodies against phospho SRC Tyr416 (1:2000; D49G4; 0004) were from Cell Signaling Technologies, against VDAC1/Porin (1:2000; EPR10852(B); 1095790-3) from Abcam (Cambridge, UK), against FAK pY397 (1:3000; 141-9; 4891961) from Invitrogen (Waltham, MA, USA) and against AKT (pan) (1:1000; C67E7; 28) and phospho-AKT Ser473 (1:1000; D9E; 27) from Cell Signaling Technologies. Primary polyclonal rabbit antibodies against phospho-p44/42 MAPK Thr202/Tyr204 (dilution: 1:1250; lot: 13) and against p44/42 MAPK (1:1250; 30) were from Cell Signaling Technologies. Secondary polyclonal murine antibodies IgG (H+L) from goat conjugated to horseradish peroxidase (1:20.000; 63332) were from Jackson Immuno Research (Cambridgeshire, UK) and against rabbit (1:10.000; 20023997) from DAKO (Glostrup, Denmark).

The 10% or 12% sodium dodecyl sulfate-polyacrylamide gel electrophoresis and immunoblot detection were performed as described in Moch et al., 2013 [18] with the following modifications. Proteins were transferred onto PVDF membranes at 100 V for 60 min using a Mini-Protean III Cell system (Bio-Rad Laboratories, Hercules, CA, USA). Membranes with bound proteins were blocked for 1 h in 10% (*v*/*v*) Roti^®^-Block (Carl Roth) and incubated with primary antibodies in buffer (50 mM Tris, 130 mM NaCl, 0.1% [*v*/*v*] Tween 20, pH 7.6) containing 1% or 2% or 10% (*v*/*v*) Roti^®^-Block. The same buffer with 1% (*v*/*v*) Roti^®^-Block was used for secondary antibody incubation. Bound antibodies were detected with Super Signal™ West Pico PLUS chemiluminescence substrate (Thermo Fisher Scientific, Waltham, MA, USA) after 5 min incubation on a Fusion-Solo.WL.4M with Fusion-Capt Advance Software 16.06 (Vilber Lourmat, Marne-la-Vallée, France). Protein mass was determined with the ProSieve QuadColor Protein Marker (Lonza, Basel, Switzerland). Unprocessed immunoblots and protein transfer controls are included in Appendix A.

### 4.5. Image Analysis and Statistical Analysis

Microscopy images were processed and analyzed in the Fiji distribution of ImageJ v.1.54 software package [107,108]. The subsequent translocation of fluorescence from the unbleached cell part to the bleached region was measured as described previously [18] with two modifications. Firstly, a smaller ROI 7.03 × 5.27 µm (80 × 60 pixel) was utilized that was within the bleached area 14.06 × 10.54 µm (160 × 120 pixel) to exclude confounding effects of unbleached GFP moving into the respective ROI during the recovery phase. Secondly, the background was measured 1 min after bleaching within the ROI and this value was subtracted from every time point. Remaining fluorescence before bleaching was defined as 100% and 1 min after bleaching as 0%. Accordingly, recovery was measured starting at time point of 1 min after bleaching, which corresponds to time-point of 0 in all plots shown in this study.

For statistical analysis of each independent FRAP experiment, the last time point (10 min) was tested for outliers (alpha = 0.05) with GraphPad online outlier calculator (GraphPad, San Diego, CA, USA; accessed in 2024) and, when applicable, the data set belonging to the farthest outlier was removed. Next, the Gaussian distribution for the last time point was tested with the D’Agostino and Pearson omnibus normality test and with the Shapiro–Wilk normality test using GraphPad 12 software (GraphPad). Since some data sets did not follow a normal distribution, we chose to use nonlinear statistical tests for final analysis. For single-drug experiments, we used a Mann–Whitney test with *p* < 0.05 as significant. When comparing a control with two drugs, we used a Kruskal–Wallis test and Dunn’s multiple comparisons post-test, considering *p* < 0.05 as significant.

Immunoblot quantification was performed with the one-dimensional electrophoretic gel analysis tool in Fiji distribution of ImageJ software. VDAC1 was used as loading standard.

Plots and fitting curves (one-phase exponential association) were made with GraphPad 10.4.1 as well. Figures were prepared with Affinity Designer 2 (Serif Ltd., Nottingham, UK). Appendix A was encoded in h.264 video format using the open-source video transcoder Handbrake v.1.8.2.

## Figures and Tables

**Figure 1 ijms-26-05476-f001:**
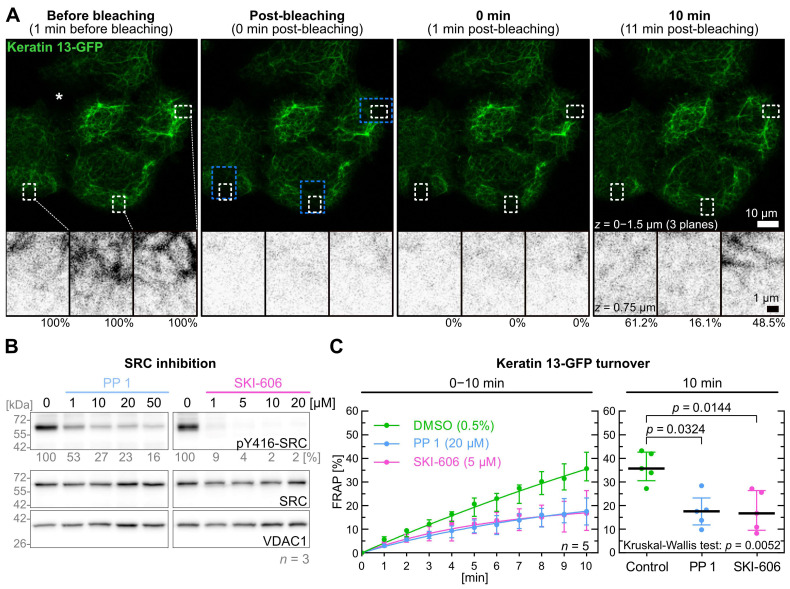
SRC inhibition results in a decrease of keratin filament turnover. Panel (**A**) presents fluorescence images of the periphery of an A431-derived AK13-1 cell colony. The images were captured during a FRAP experiment in which AK13-1 cells were preincubated for 5–10 min with 0.5% DMSO. The maximum-intensity projections show keratin 13-GFP signals that were compiled from the lower three focal planes before bleaching, immediately after bleaching and at two later time points. The black and white images below are high magnifications of single confocal planes of the regions demarcated by white broken lines that were used for measurements and located within the bleached areas (demarcated by blue broken lines). The measured relative signal intensity values are provided underneath. The asterisk in the top left micrograph is positioned in a region of the colony containing two cells that do not express detectable levels of keratin 13-GFP. (**B**) shows representative immunoblots of AK13-1 cells treated with two structurally different SRC inhibitors (i.e., PP1 and SKI-606) at increasing concentrations for 5 min. The relative intensity values of the Y416-SRC phosphorylation immunosignals are provided below the top panel (*n* = 3; percentage of phosphorylated SRC versus VDAC1). (**C**) shows the time-dependent fluorescence recovery graphs of keratin 13-GFP after bleaching in response to 0.5% DMSO alone (control), 20 µM PP1 and 5 µM SKI-606 after 5–10 min pretreatment (*n* = 5; 55–58 cells per condition). The FRAP experiments were performed as illustrated in (**A**). For clarity, the plot at left shows only one median per time point for each condition. The plot on the right presents the fluorescence recovery at the 10-min time point providing the median results of each independent experiment as a dot. Whiskers correspond to interquartile ranges in both diagrams; statistical analysis was performed with a Kruskal–Wallis test at 10 min with a Dunn’s multiple comparisons post-test.

**Figure 2 ijms-26-05476-f002:**
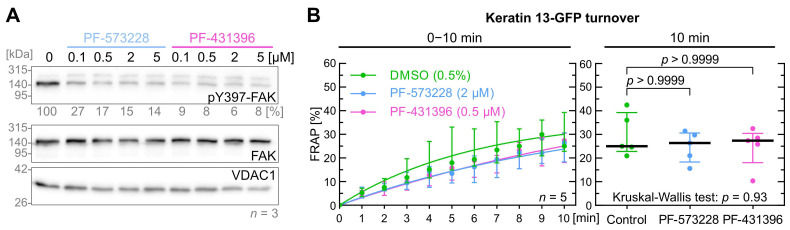
FAK inhibition does not affect keratin filament turnover. (**A**) shows representative immunoblots of AK13-1 cells treated with two structurally different FAK inhibitors (i.e., PF-573228 and PF-431396) at increasing concentrations for 5 min. The relative intensity values of the Y397-FAK phosphorylation immunosignals are provided below the top panel (*n* = 3; percentage of phosphorylated FAK versus VDAC1). (**B**) shows the time-dependent fluorescence recovery graphs of keratin 13-GFP after bleaching in response to 0.5% DMSO (control), 2 µM PF-573228 and 0.5 µM PF-431396 (*n* = 5; 41–45 cells per condition). The FRAP experiments were performed as illustrated in Figure 1A. The plot on the right presents the fluorescence recovery at the 10-min time point, depicting the median results of each independent experiment as a dot. Whiskers correspond to interquartile ranges in both diagrams; statistical analysis was performed with a Kruskal–Wallis test at 10 min with a Dunn’s multiple comparisons post-test.

**Figure 3 ijms-26-05476-f003:**
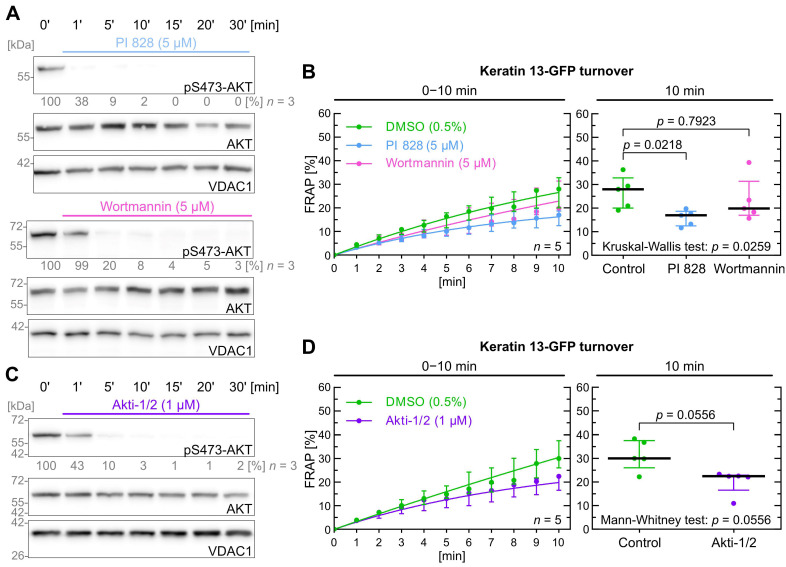
PI3K and AKT inhibition do not significantly affect keratin filament turnover. (**A**) shows representative immunoblots of AK13-1 cells treated with two structurally different PI3K inhibitors (i.e., 5 µM PI 828 and 5 µM wortmannin) in a time-dependent manner. The relative intensity values of S473-AKT phosphorylation are provided below the top panel (*n* = 3; percentage of phosphorylated AKT versus VDAC1). In this case we determined PI3K inhibition based on downstream AKT inhibition. (**B**) shows the time-dependent fluorescence recovery curves of keratin 13-GFP after bleaching in response to 0.5% DMSO (control), 5 µM PI 828 and 5 µM wortmannin (*n* = 5; 57–59 cells per condition). The FRAP experiments were performed as illustrated in Figure 1A. The plot on the right presents the fluorescence recovery at the 10-min time point depicting the median results of each independent experiment as a dot. Whiskers correspond to interquartile ranges in both diagrams; statistical analysis was performed with a Kruskal–Wallis test at 10 min with a Dunn’s multiple comparisons post-test. (**C**) shows representative immunoblots of AK13-1 cells treated with the dual AKT1 and AKT2 inhibitor Akti-1/2 (1 µM) in a time-dependent manner. The relative intensity values of phosphorylated S473-AKT are provided below the top panel (*n* = 3; percentage of phosphorylated AKT versus VDAC1). (**D**) shows the time-dependent fluorescence recovery curves of keratin 13-GFP after bleaching in response to 0.5% DMSO (control) and 1 µM Akti-1/2 (*n* = 5; 100–103 cells per condition). Whiskers correspond to interquartile ranges in both diagrams; statistical analysis was performed with a Mann–Whitney test at 10 min.

**Figure 4 ijms-26-05476-f004:**
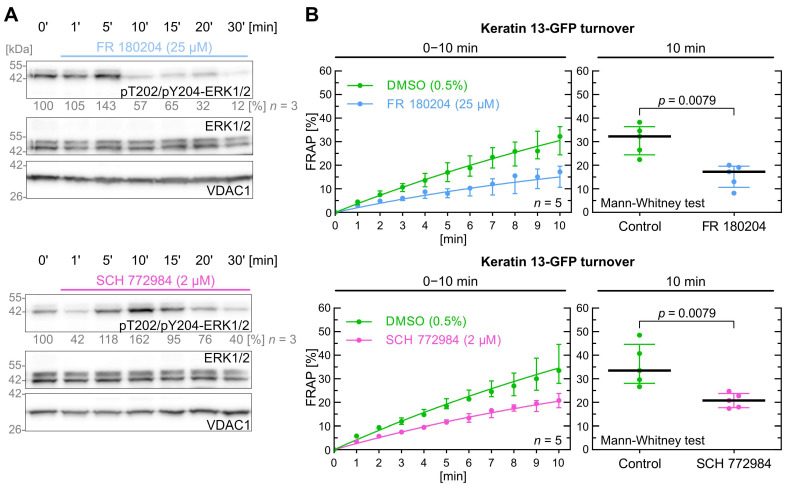
ERK1/2 inhibition decreases keratin turnover. (**A**) shows representative immunoblots of AK13-1 cells treated with two structurally different PI3K inhibitors (i.e., 25 µM FR 180204 and 2 µM SCH 772984) for different periods. The relative intensity values of phosphorylated T202/Y204-ERK1/2 are provided below the top panel (*n* = 3; percentage of phosphorylated AKT versus VDAC1). (**B**) shows the time-dependent fluorescence recovery graphs of keratin 13-GFP after bleaching in response to 0.5% DMSO (control), 25 µM FR 180204 and 2 µM SCH 772984 in two independent sets of experiments (*n* = 5; 102–105 cells per condition). The FRAP experiments were performed as illustrated in Figure 1A but with an extended 10–15 min drug pretreatment. The plot on the right presents the fluorescence recovery at the 10-min time point depicting the median results of each independent experiment as a dot. Whiskers correspond to interquartile ranges in both diagrams; statistical analysis was performed with a Mann–Whitney test at 10 min.

**Table 1 ijms-26-05476-t001:** Impact of kinase inhibition on keratin turnover.

Kinase	Inhibitor	Inhibitor [µM]	Keratin Turnover ^1^
SRC	PP 1	20	↓ 51% (*p* = 0.032)
	SKI-606	5	↓ 53% (*p* = 0.014)
ERK1/2	FR 180204	25	↓ 47% (*p* = 0.008)
	SCH 772984	2	↓ 37% (*p* = 0.008)
FAK	PF-573228	2	↑ 6% (*p* > 0.999)
	PF-431396	0.5	↑ 10% (*p* > 0.999)
PI3K	PI 828	5	↓ 39% (*p* = 0.022)
	Wortmannin	5	↓ 29% (*p* = 0.792)
AKT	Akti-1/2	1	↓ 25% (*p* = 0.056)

^1^ Change in keratin turnover after 10 min.

## Data Availability

All statistical data and immunoblots are provided in the Appendix A. Unprocessed confocal recordings in LSM5 format can be provided upon personal request.

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
