# Peer review of "SRC and ERK Regulate the Turnover of Cytoskeletal Keratin Filaments"

_ijms, 2025, doi:10.3390/ijms26125476_

Round 1
Reviewer 1 Report
Comments and Suggestions for Authors
The article, “EGFR signaling impacts cytoskeletal keratin filament turnover through Src and ERK” by Moch and Leube reports the involvement of SRC and ERK in the regulation of ketatin filament turnover in response to EGF, while PI3K/AKT and FAK did not. With pharmacological inhibitors, they demonstrated differential involvement of EGFR downstream signaling pathways in the keratin turnover. Overall, the experimental approach is straight forward and well performed. However, further studies, mentioned in the Major comments, are recommended to perform before publication.
Major comments
- Authors discussed the phosphorylations of keratin by SRC and ERK1/2 in lines 196 – 298. Why don’t you check the phosphorylation of keratin when the cells were treated with each kinase inhibitors? I understand that it will be an extreme job, but very informative.
- Why don’t you check the cells movement or motility in the presence of kinase inhibitors? Is there any effects on the cells’ movement or motility or other behavior in the presence of each inhibitor?
Minor comments
- There are no information on authors’ affiliation.
- The term “Src” should be “SRC”. For protein & gene name, please use official name. Please refer UniProt (uniprot.org) and HGNC (https://pmc.ncbi.nlm.nih.gov/articles/PMC7494048/).
- There are no descriptions on ERK, PI3K, FAK, etc. Abbreviations should be described by their full name at the first appearance in the manuscript. Please make sure for others in the entire manuscript.
- In Figure 1C, please provide information for number (n) of analysis. In line 105, you described as “each experiment series was performed on five different days”. Does this mean n = 5?
- Please put the reference for the description in line 114.
- The Original Images are hard to understand. There are no information on each lane and size markers. In addition, the images are inversed images. Authors should provide more information on the Original images.
- Please, provide references for SRC inhibitors and FAK inhibitors described in line 114 and line 156, respectively. References for other inhibitors should also be provided properly.
<The End>
Author Response
Please see the attachment.
The document includes the responses to the Editor, both Reviewers, and a marked-up version of the manuscript with all corrections.

Reviewer 2 Report
Comments and Suggestions for Authors
Moch and Leube examined what signaling cascades are important for keratin filament turnover. They found that Src and ERKs are more significant factors. Although the results might be informative for researchers studying organization of intermediate filament cytoskeleton, there are several concerns.
Comments:
1) Although the authors emphasized EGFR signaling, they did not employ EGF stimulation in this study. In this regard, EGF stimulation was employed in their previous study (ref. 18). FBS includes various extracellular factors, which activate different signaling cascades including EGFR and ERKs. Thus, to specify EGFR signaling, EGF stimulation is essential.
2) To inhibit target kinase activity, the authors used high concentrations of inhibitors throughout the manuscript. Such high concentration may affect the activities of other related kinases non-specifically. In this regard, ERK inhibitors first activate and then decrease ERK activity (Figure S1), which may indicate side effects (non-specificity of the inhibitors). To confirm the involvement of Src and ERKs, knock-down experiments are essential.
3) A schematic diagram about how Src and ERKs are involved in keratin filament turnover should be shown in a figure along with discussion.
4) It is not clear why keratin-13 is selected as a representative keratin in this study. Is it simply because the keratin-13-EGF cell line can be used? It is not clear importance of keratin-13 in keratin intermediate filament cytoskeleton. Can the results of keratin-13 be generalized as those of “keratin”?
Author Response

(The authors gave the same response as above.)

Round 2
Reviewer 2 Report
Comments and Suggestions for Authors
Moch and Leube responded to my comments and revised the manuscript. However, I recommend additional amendments.
Comments:
1) Although the authors modified the title, the main text still includes a lot of the descriptions related to “EGFR signaling”, which confuses readers. Again, FBS includes various extracellular factors, which activate different signaling cascades involving the kinases listed in the new Table 1. The EGFR signaling cascade is just one of them. Thus, re-writing without focusing on the EGFR signaling cascade is essential. If it is impossible, experiments with EGF stimulation are essential.
2) Please add information on the concentration of the inhibitors in the new Table 1, which helps interpretation by readers.
3) The response “Keratin 13 is one of the major keratin polypeptides in A431 cells. We have previously shown that it labels all keratin filaments in these cells because of its ability to pair with different partner polypeptides (Windoffer and Leube, 1999, JCS 112: 4521). As a result, we and others have used the AK13-1 cell line in multiple studies.” is informative for readers’ understanding. So, please include this information in the Introduction section.
Round 3
Reviewer 2 Report
Comments and Suggestions for Authors
Moch and Leube added information which helps potential readers’ understanding.